

# ELI: an IoT-aware big data pipeline with data curation and data quality

Francisco José de Haro-Olmo[1], Alvaro Valencia-Parra[2], Ángel Jesús Varela-Vaca[2], José Antonio Álvarez-Bermejo[1] and María Teresa Gómez-López[2]

[1] Departamento de Informática, Universidad de Almería, Almería, Spain
[2] Departamento de Lenguajes y Sistemas Informáticos, Universidad de Sevilla, Sevilla, Spain

## ABSTRACT

The complexity of analysing data from IoT sensors requires the use of Big Data technologies, posing challenges such as data curation and data quality assessment. Not facing both aspects potentially can lead to erroneous decision-making (*i.e.*, processing incorrectly treated data, introducing errors into processes, causing damage or increasing costs). This article presents ELI, an IoT-based Big Data pipeline for developing a data curation process and assessing the usability of data collected by IoT sensors in both offline and online scenarios. We propose the use of a pipeline that integrates data transformation and integration tools and a customisable decision model based on the Decision Model and Notation (DMN) to evaluate the data quality. Our study emphasises the importance of data curation and quality to integrate IoT information by identifying and discarding low-quality data that obstruct meaningful insights and introduce errors in decision making. We evaluated our approach in a smart farm scenario using agricultural humidity and temperature data collected from various types of sensors. Moreover, the proposed model exhibited consistent results in offline and online (stream data) scenarios. In addition, a performance evaluation has been developed, demonstrating its effectiveness. In summary, this article contributes to the development of a usable and effective IoT-based Big Data pipeline with data curation capabilities and assessing data usability in both online and offline scenarios. Additionally, it introduces customisable decision models for measuring data quality across multiple dimensions.

Corresponding author
Francisco José de Haro-Olmo,
fdo730@inlumine.ual.es

## INTRODUCTION

The present and the near future are full of scenarios in which devices of the Internet of Things (IoT) are equipped with sensors to gather a myriad of data from various sources to carry out business intelligence processes or decision-making actions subsequently. These devices can be found not only in the industrial environment, but also increasingly in the everyday scenario of each of us. These interconnected devices, such as smart appliances, can collect large chunks of data from their environment that are later injected into Big Data pipelines. In *Ceravolo et al. (2018)*, the authors propose a sequence of stages to follow to achieve the business intelligence-based goals planned using Big Data technologies. However, several issues are raised regarding the data collected by IoT devices through

their onboard sensors and how these data will be processed into Big Data systems. Some of the main challenges when IoT data are incorporated in a Big Data system are (1) the heterogeneity of data sources, (2) the processing of stream data along with its gathering process, and (3) the collection of large chunks of data that can increase issues with data quality, privacy, and security, *Choi et al. (2017)*. In this article, we focused on the extension of a Big Data pipeline incorporating both data curation and quality phases adapted to IoT information. This is not a straightforward aspect; rather, it is a complex issue because the evaluation is highly context-dependent (*i.e.,* the origin of the data and its nature, the semantics, as well as the problem domain). Low-quality data (*i.e.,* incomplete data, wrong readings, null readings) will lead analytical processes to unintended consequences; *Rahm & Do (2000)*, *Redman (1998)* and *Hernández & Stolfo (1998)* resulting in bad decisions recommended by processing incorrectly treated data, introducing errors into processes, causing damage or increasing costs. Therefore, it is critical to incorporate quality methods that ensure a proper data acquisition process and prevent low-quality data from being processed by enforcing a minimal level of usability in the scenario and context in which it is carried out. However, one of the challenges is that the meaning of quality depends on the context. For this reason, any solution that supports data curation and quality must provide mechanisms to describe business rules according to the required domain in each case.

The challenges to be taken into account are the following: (1) managing the heterogeneity of data coming from IoT sensors in offline or online environments; (2) developing an adaptable mechanism to provide data curation including data quality assessment depending on the context. To face these challenges, three objectives have been defined: (i) to propose an IoT-aware Big Data pipeline that allows data curation and data quality from different IoT sensor devices according to the context; (ii) the definition of mechanisms that enable data transformation to integrate them into a single data set; (iii) the integration of mechanisms based on context-aware decision models that facilitate the automated measurement and evaluation of data quality, and; (iv) the evaluation of the current proposal through a real scenario from the agricultural context by analysing the results obtained. One example of this type of IoT context can be found in the agribusiness sector. There is a catalyst in the agribusiness industries' move to a model based on automated data collection and the creation of decision processes that rely on those data. Therefore, the concept of a smart farm is increasingly being adopted, as mentioned in *Kosior (2018)*. Figure 1 depicts a classical smart farm scenario. The nature of the sources from where data is gathered makes it more than necessary to undergo preprocessing stages before feeding the pipeline that runs the Big Data system. Furthermore, the hostile environment in which these devices operate (high temperatures, humidity, battery-operated devices, radio connections, *etc.*) is an added handicap to ensure that data maintain quality. The fact that data sources are diverse and heterogeneous and the complexity of their sources can raise serious issues that can negatively impact data quality, *Ilyas & Chu (2019)* (*e.g.*, missing or incomplete data due to power supply problems). This leads to taking into account that the data collected by IoT sensors, before their integration into the Big Data pipeline, must be preprocessed (*i.e.,* data curation) and also must be submitted to an evaluation to determine their coherence. Thus, data that do not meet the requirements need to be discarded or improved; only data

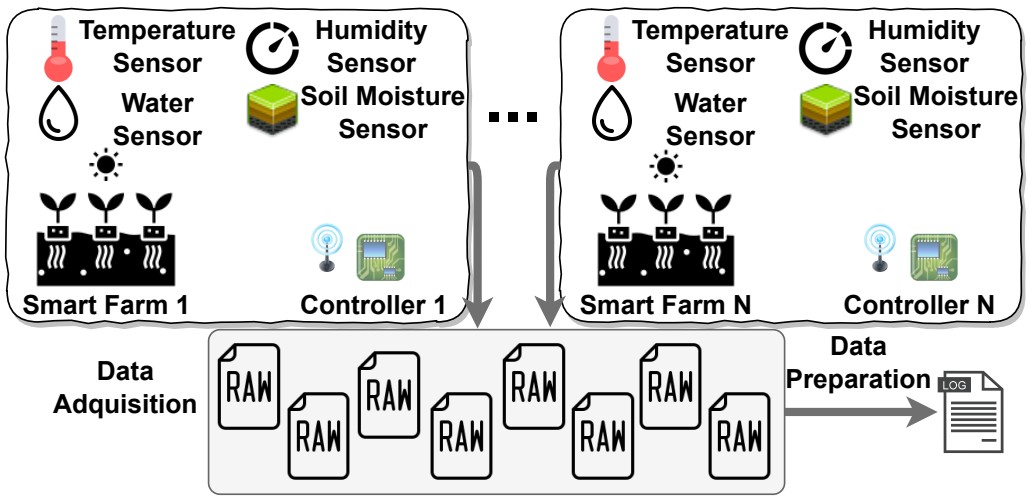

**Figure 1** Smart Farm IoT context example.

that meet quality thresholds are considered to add value to the system. This will have an impact on the subsequent processes that depend on the data collected.

For tackling the objectives, we integrate into a Big Data pipeline the data curation stage using CHAMALEON (*Valencia-Parra et al., 2019*) to integrate data coming from heterogeneous data sources, and the DMN4DQ (*Valencia-Parra et al., 2021*; *de Haro-Olmo et al., 2021*) to support a data quality assessment. Neither CHAMALEON nor DMN4DQ have been applied in IoT environments, at the same time that they have not been integrated into the same Big Data pipeline. For this reason, ELI has been proposed as an applicable pipeline that provides a framework for testing and analysing results in offline and online scenarios. We aim to achieve data curation and quality for the context at hand by proposing an approach that allows users to define a standard, easy-to-use and executable mechanism for specifying conditions in terms of what dimensions, what restrictions, *etc*. In summary, we have a publication that deals with data quality assessment, *Valencia-Parra et al. (2021)* and we have another article about data curation, *Valencia-Parra et al. (2019)* and *de Haro-Olmo et al. (2021)*, respectively. In this new contribution we define a unified Big Data pipeline with tools for data quality assessment and data curation for sensorized environments, for which we have integrated CHAMALEON for curation and DMN4DQ for data quality assessment. This has not been done in any other contribution we have published before. The main contribution of this work is the provision of the framework and the evaluation that has been done under offline and online scenarios.

The rest of the article is organised as follows. In the following section, we collect work related to IoT technology and Big Data pipelines, including the study of tasks related to the data acquisition and curation process. Section 'ELI: IOT-Aware Big Data Pipeline Approach' describes ELI, the proposed model for integrating data collected through IoT sensors and the Big Data pipeline. Section 'DMN4DQ in a nutshell' briefly introduces the data quality approach that is integrated into the Big Data pipeline. In Section 'Case Study:

the agri-food case' a concrete case is shown with data coming from an agricultural farm and in which the ELI pipeline in this work is applied. In Section 'Evaluation and Analysis' the results of applying the model to the case study are analysed. In Subsection 'Discussion', we contrast the results obtained between the different offline and online scenarios. In the final part, 'Conclusions and Future Work', conclusions and proposals for future work are drawn.

## RELATED WORK

Faced with the challenge of integrating IoT technology with Big Data, several studies have been carried out, with relevant contributions to the architecture of the proposed models (*Cecchinel et al., 2014*; *Marjani et al., 2017*; *Taherkordi, Eliassen & Horn, 2017*). In *Cecchinel et al. (2014)* an architecture composed of boards that integrate IoT sensors is proposed that can intercommunicate through bridges established between different groups of sensors, which in turn would be responsible for sending the data to the cloud. Extending the perspective, *Marjani et al. (2017)* offers a somewhat more complex architecture, incorporating, in addition to IoT sensors, other elements with distinct functions for data collection through IoT sensors, networks connecting the various sensors, and providing a gateway for storing data in the cloud. Data are subsequently analysed in Big Data processes and add value to the entire process. As far as real-time data processing is concerned, the proposals shown so far do not show significant differences. *Taherkordi, Eliassen & Horn (2017)* specifies the architecture and component elements for a Big Data system to be able to process data from IoT environments. Several authors have proposed a lambda architecture for real-time processing of data collected through IoT sensors, *Villari et al. (2014)* and *Kiran et al. (2015)*. However, these works are focused on defining or processing aspects of the architecture, ignoring the data quality and curation aspects.

Data curation is one of the key issues that must be addressed for data collected by sensor devices in IoT environments to be valid and usable in the system. *Freitas & Curry (2016)* analyse the different challenges of Big Data with regard to the integration of curation and how new technologies deal with these new requirements. An important task is data acquisition (*Lyko, Nitzschke & Ngonga Ngomo, 2016*) and how new technologies have to deal with different problems and challenges (*Lyko, Nitzschke & Ngonga Ngomo, 2016*). We found other articles, such as *Stonebraker et al. (2013)*, *Choi et al. (2017)*, *Lyko, Nitzschke & Ngonga Ngomo (2016)*, *Beheshti et al. (2017)*, *Rehm et al. (2020)*, *Yang et al. (2017)* and *Murray et al. (2020)* that address the tasks that are carried out to make the data reach a minimum quality for it to be used correctly. These tasks include data cleansing, data transformation, or data formatting, not forgetting that data may originate from heterogeneous devices with different ways of representing data, on which necessary transformations must be performed to obtain a set of resulting data that are compatible with each other.

The proposal we carry out can provide an improvement to the architecture proposed in *Yang et al. (2017)* in which the importance of having accurate data is highlighted as a previous step to cloud storage. In this way, the proposed architecture takes into account the process of treatment of the acquired data, such as error detection, and refers to the

lack of new techniques to achieve this objective. Our work introduces as a novelty the use of the DMN4DQ in a Big Data pipeline combined with data curation process, where it allows the definition of business rules under the DMN notation (*Object Management Group, 2019*), which are applied to the received data in order to measure and evaluate the quality of the data and thus accept or reject each data, reporting on the quality of each data separately. Decision model and notation (DMN) is a modelling language and notation for the specification of business rules, which provides engines for automatic decision-making.

Several tasks make up the process of data curation, reading, collection, classification, and enrichment, which must be perfectly defined and coordinated so that the result is adequate (*Beheshti et al., 2017*), also addressing the proposal of an API to facilitate the process of data curation for users.

In *Rehm et al. (2020)*, to avoid poor quality data in text documents, a platform is proposed to organise the data curation workflow in different contexts. To this end, they focus on aspects such as ensuring the pattern that data follows, incorporating machine learning techniques to assess data quality, and increasing the credibility of the data by making use of reference sources.

A very interesting proposal is the one made by *Murray et al. (2020)*, which deals with a tool in the field of COVID-19 disease, which deals with the detection of symptoms in various patients and on which data curation techniques are applied, as the data originally acquired does not always have the desired quality. To this end, a data cleaning and curation process is proposed that involves processing the data to obtain the appropriate format and types of data in each field. The data quality analyse process proposed in our work can contribute positively to the improvement of the process, complementing its data curation proposal and, by providing business rules for data quality, can provide measurement and evaluation of the data incorporated into the system, including in terms of usability.

This is an innovative approach to the work we propose, as it incorporates a proven framework in the field of data quality that allows data to be tracked throughout its lifecycle, systematically applying business rules according to the context in which it is applied. It is important to note that this approach can be applied to any context that makes use of IoT environments that need to collect data from sensors for automatic processing and with minimum quality requirements.

## ELI: IOT-AWARE BIG DATA PIPELINE APPROACH

The increasing complexity of sources and the later analysis has made it necessary to develop a set of activities to acquire, prepare, analyse and interpret the data. This set of activities is choreographed in the known as Big Data pipelines to facilitate the data flow when data have variety, velocity, and an important volume. These three Vs, known as relevant features of Big Data, tend to appear in IoT environments. IoTs describe a network of physical objects to extract data from the real environment to interchange data among devices *via* the Internet. Previous proposals have incorporated the management of complex semi-structured data, including data curation (*Valencia-Parra et al., 2019*). However, online data quality analysis and application in an IoT environment were out of the scope of the previous proposal.

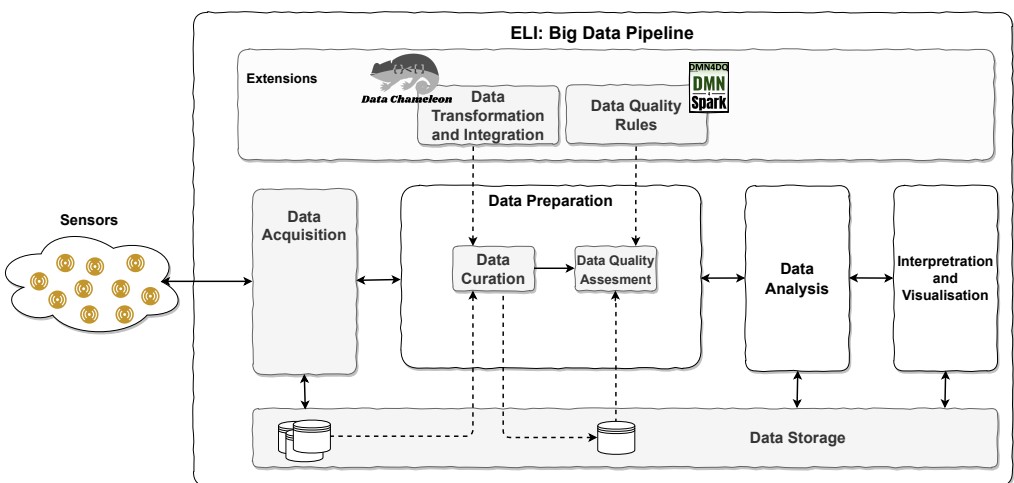

**Figure 2** ELI pipeline overview.

To extract valuable data from IoT sensor readings, several tasks must be performed in a coordinated manner to form a Big Data Pipeline (*Ceravolo et al., 2018*; *Pääkkönen & Pakkala, 2015*; *Curry, 2016*; *Ardagna et al., 2017*). Building on existing models, this leads to the proposal of ELI, an IoT-aware Big Data Pipeline model, and it is shown in Fig. 2.

It is important to remark on the differences between machine learning and Big data pipelines. A machine learning pipeline aims to produce a machine learning model, but a Big Data pipeline aims to extract value from data (*Valencia-Parra, 2022*; *Ceravolo et al., 2018*). The current machine learning processes or pipelines are based on data preparation, model training and evaluation, and returning a trained model. Big data pipelines can integrate a business intelligence or machine learning process in the data analysis stage to extract value from data.

In the proposed model (see Fig. 2), after the collection of data from IoT sensors, a pipeline is established to carry out the following tasks: (1) Data Acquisition of sensors and arrange them in a situation to be used in the next phase of the pipeline; (2) Data Preparation, which involves cleaning and formatting operations, and integrating of the original data to adapt them to the appropriate format to be processed in the next phase; according to *Curry (2016)*, this task can be added to the following one; (3) Data Analysis, where valuable information is obtained from the data by applying business intelligence, *Ardagna et al. (2017)* and data science, *Poornima & Pushpalatha (2016)*; *Marjani et al. (2017)*; (4) Interpretation and Visualisation, the final phase of the pipeline in which to display the valuable information that emerges after the entire Big Data pipeline process. (5) Data Storage, whose main objective is to store data from the moment of acquisition so that it is available throughout the entire process.

In addition to these tasks along the pipeline, two more tasks are introduced sequentially, Data Curation and Data Quality Assessment. Data Curation is defined as "the act of discovering a data source of interest, cleaning and transforming the new data, semantically integrating it with other local data sources, and de-duplicating the resulting composite",

*Stonebraker et al. (2013)*. It consists of preparing the data in the form that is required in the system to be properly processed and to be able to obtain quality data. Data Quality Assessment: *Batini & Scannapieco (2016)* evaluates the data using rules on various dimensions of data quality to determine the degree of quality of each piece of data fed into the system.

To achieve the inclusion of data curation and data quality assessment, it is proposed to use CHAMALEON (*Valencia-Parra et al., 2019*) and DMN4DQ, *Valencia-Parra et al. (2021)*, respectively.

## CHAMALEON in a nutshell

Data transformation has been an important topic in the data-warehouse area and currently applied to Big Data environments that need new mechanisms and tools to combine huge quantities of heterogeneous information. The data transformation process is a difficult, error-prone, time-consuming, but crucial stage of data analytics. In order to facilitate advanced data transformations in Big Data environments, CHAMALEON (https://github.com/IDEA-Research-Group/Data-Chameleon) (*Valencia-Parra et al., 2019*) proposes a set of complex functions in a concise syntax supported by a Domain-Specific Language (DSL). CHAMALEON was tested on a Big Data framework to obtain an initial benchmark about the impact on performances of size factors such as the number and the size of tuples.

CHAMALEON enables data curation of complex heterogeneous data sources using a DSL to facilitate data wrangling. The DSL provided by CHAMALEON allows transformation using templates with operations, for example, to change schemas. The data from sensors are ingested and stored in different data stores (*i.e.,* cvs or txt files) in the data storage layer. Afterwards, the data are curated using the CHAMALEON template, generating a single dataset. After the data curation process, DMN4DQ is used to assess the usability of data quality rules.

Applied to IoT environments, CHAMALEON permits the definition of a set of transformations to obtain a single repository where data quality assessment and later analysis are applied, as shown in Fig. 3. This is the first integration of CHAMALEON into a Big data pipeline to curate data from IoT environments.

## DMN4DQ in a nutshell

DMN4DQ method (*Valencia-Parra et al., 2021*) is designed to automatically obtain an evaluation of the quality of the data and its usability in the context to which it belongs and according to predefined rules. As shown in Fig. 4, it is a matter of defining the different rules that will make it possible to determine the quality of the data in terms of usability at different hierarchical levels.

To define the business rules, DMN4DQ is based on Decision Model and Notation (DMN) (*Object Management Group, 2019*), a modelling language and notation for the specification of business rules, providing engines for automatic decision-making. This notation can be used to represent the business logic that will be applied to the data to be

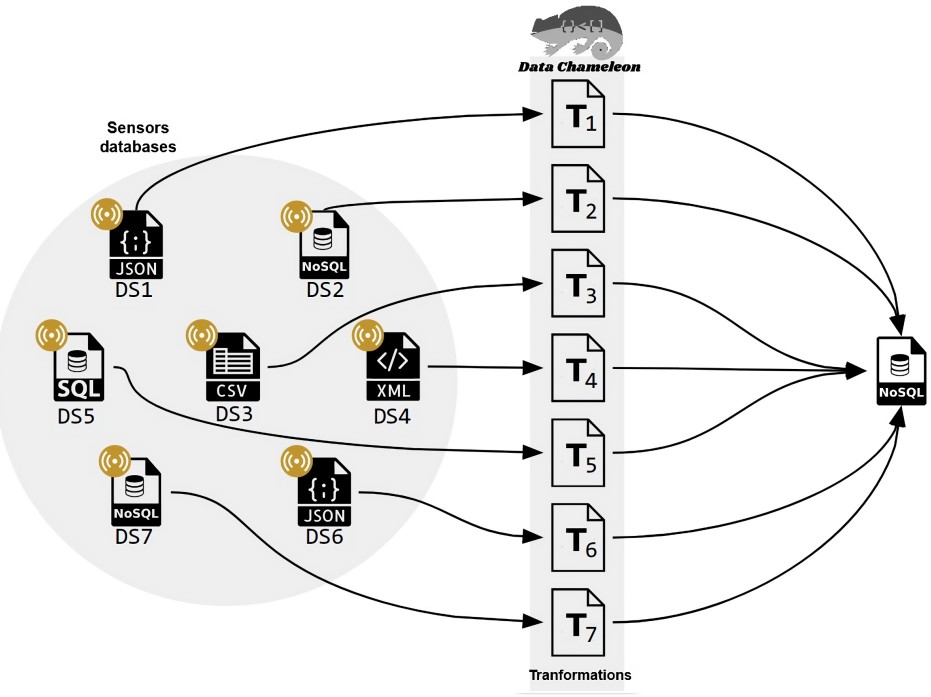

**Figure 3** Data preparation for an IoT scenario with multiple data sources.

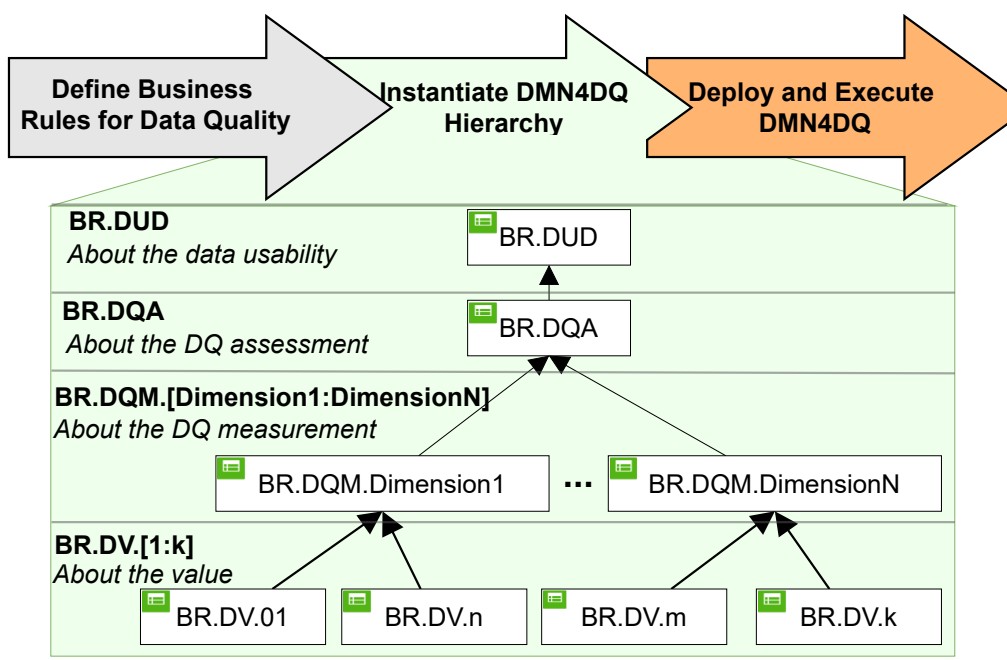

**Figure 4** DMN4DQ method.

| | Input | | Output |
|---|---|---|---|
| F | **Completeness** | **Accuracy** | **DQA** |
| | Number | Number | {"suitable quality", "sufficient quality", "bad quality", "non usable"} |
| 1 | >=5 | >=100 | "suitable quality" |
| 2 | >=3 | >=60 | "sufficient quality" |
| 3 | >=1 | <60 | "bad quality" |
| 4 | - | - | "non usable" |

**Figure 5   DMN table example.**

processed, achieving a classification of the data incorporated into a business process as evaluated by one of the existing engines.

These rules are based on existing "if-then" statements in programming languages. We can see an example of a decision rule definition in Fig. 5 implemented through a table, in which the data that make up the input for the rule are defined, arranging the different conditions in different numbered rows, conditions are expressed in the FEEL language (*Object Management Group, 2019*). Each rule will have an output that will be activated depending on the input conditions that are met. Both input and output data types can be string, integer, decimal, date, and Boolean.

Following the example in Fig. 5, those data submitted to the input whose value, for instance, in the first entry, for the *Completeness* attribute must be greater than or equal to 5 and the entries for the *Accuracy* attribute must be greater than or equal to 100. In this case, if the rule is satisfied, you would get an output, for instance, DQA (Data Quality Assessment) with the assigned value of "suitable quality". In the example of Fig. 5, the decision about the 'DQA' is taken according to the values of the inputs ⟨⟩'Completeness', 'Accuracy' ⟨⟩. The evaluation of each rule is executed and ordered from the first row to the last. For example, for a tuple of input values ⟨⟩3.0, 150 ⟨⟩, the output is 'sufficient quality' since row 1 is not satisfied, but row 2 is. Row 3 is not evaluated since inputs have matched row 2.

Likewise, the DMN model also incorporates a way to handle multiple matches in a rule, making it possible to enrich the outputs to respond to a wide variety of possible cases. For this purpose, it has a hit policy indicator, which can take the following values: unique (U), where only one rule can be activated, and it is not possible to activate any other rule in the same set; any (A), where it is possible that several rules can be activated and will be taken into account to make the output of the rule. Priority (P), in this case, although several rules can be activated by fulfilling the condition, only the one with the highest priority will be the one that is finally activated, giving rise to the output associated with the selected rule; first (F), among all the rules defined along the different rows, the first one that meets the requirements will be selected, giving as output the value associated with the output of

that rule, and; collect (C), is interesting when you want to make possible the activation of several rules and that the output is related to the aggregation of all the selected rules.

DMN tables can be incorporated in a hierarchy, where the outputs of a table can be the input in another decision, as used in the DMN4DQ method. DMN4DQ method is formed of three main phases: define the business rules for data quality according to the process requirement, instantiate the DMN4DQ to represent the defined business rules for data quality, and deploy and execute the business rules to assess the data quality.

Looking at Fig. 4, we can see that the DMN4DQ method is based on a hierarchy of DMN decision rules.

At the first level of the hierarchy, the business rules for the validation of each data are defined. At a second level, measurement of the desired data quality dimensions (*i.e.,* completeness and accuracy) is carried out, which will be taken into account at the next level of the hierarchy, where the evaluation of the data quality according to the business rules is carried out. This will conclude with a recommendation to take the data into account or to reject them if they do not exceed the minimum required quality. All this is implemented thanks to the decision model and the notation paradigm (DMN), *Object Management Group (2019)*. The names of the rules in each level are:

 (i)  BR.DV represents the business rules that are applied to each data to check that it meets the data requirements for each data type.

 (ii)  BR.DQM deals with the measurement of data quality using business rules applied for the different dimensions of data quality.

(iii)  BR.DQA integrates the business rules that perform the data quality assessment. Perform this task, it will rely on the data quality measurement performed at the previous level.

(iv)  BR.DUD is the final part of the decision rule hierarchy, where business rules are applied that lead to the determination of the usability of the data, and where it determines whether to use the data or reject them.

This method can be implemented to handle large datasets using DMN4Spark. This is a library that is compatible with the Scala programming language and makes it possible for users to use the Camunda DMN engine in Big Data environments together with Apache Spark.

Therefore, the DMN4DQ enables the implementation of a software architecture based on commercial reference implementations, with the aim of automating, through the application of business rules represented by DMN notation, the process of generating usage recommendations on the data collected by IoT sensors.

## CASE STUDY: THE AGRI-FOOD CASE

To carry out this work, a real case study has been carried out using ELI to incorporate data from an agricultural farm, captured by different IoT sensors. The study aims to evaluate the quality of the data, specifically the usability, so that we can dispense with data of insufficient quality. To achieve this goal, ELI is used to define the various phases related to curation and quality processes.

### Define data context and describe the dataset

Through 42 IoT sensors located in different places, data is collected from an agricultural farm, covering humidity, soil temperature, and electrical conductivity, all taken at five different depths (30, 60, 90, 120, and 150 cm). The information collected is stored in plain text files, taking one reading every hour for each sensor and depth.

The case study is carried out with a database (*Gasch et al., 2017*) consisting of 3,373,699 records, where each record contains the following data:

- Location: contains the identification of the sensor (*e.g.*, CAF003) taking the reading. Dates (mm/dd/yyyy), and times (hh: mm) at which the data is collected.
- VW_30 cm, VW_60 cm, VW_90 cm, VW_120 cm, and VW_150 cm representing the humidity ($m^3/m^3$) detected at different depths, using NA as missing data.
- T_30 cm, T_60 cm, T_90 cm, T_120 cm, and T_150 cm the data associated with the temperatures ($C$, Celsius) detected at the different depths of each IoT sensor, using NA as missing data.

### Identify data quality dimensions and define business rules for data values (BR.DV)

Data quality rules are described in Fig. 6 and applied after the data curate process, which applies the quality dimensions of completeness and accuracy, dimensions included in the DMN model. At the first level of the DMN4DQ model, we find the business rules for data quality for the dimensions of completeness and accuracy. In our case, both completeness and accuracy are evaluated based on the knowledge expect extracted from the values classified as realistic collected for each IoT sensor determined by the experts. The BRs are:

1. **Completeness.** The business rules from BR.DV.01 to BR.DV08 aim to ensure that each piece of data to be processed is complete and that no part of the information has been lost. Thus, rules BR.DV.01 to BR.DV.03 ensure that data corresponding to location, date and time contain data and these are not null, being the output value assigned to false when it occurs and true in any other cases. The next business rules, BR.DV.04 to BR.DV.08, aim to ensure that readings taken by sensors at a particular depth contain a value and are not the empty string or a null value. Taking as an example the rule BR.DV.04 in Fig. 7, we can see that the first rule that is fulfilled will be activated (F:First), in such a way that if for a depth of 30 cm the readings of humidity (VW_30 cm) and the temperature at 30 cm (T_30 cm) are null, the output value 0 will be assigned; if on the other hand, only one of the readings is empty, null, or without an assigned value (NA), the output value 1 will be assigned; in any other case, the value 2 will be assigned, which is compatible with the fact that both readings contain a value, as can be seen in Fig. 7. Using these assigned output values, the quality measurement of each data can be carried out at the next higher level of the DMN4DQ model.

2. **Accuracy.** In this area of data quality validation, the aim is to ensure that the collected data are reliable and do not present unusual or out-of-range values. In the case under study, rules BR.DV.09 to BR.DV.13 are intended to classify, according to the humidity and temperature values at each depth, offering as output data the values: "realistic" if the humidity is in the range between 0.150 and 0.700 and the temperature between 1

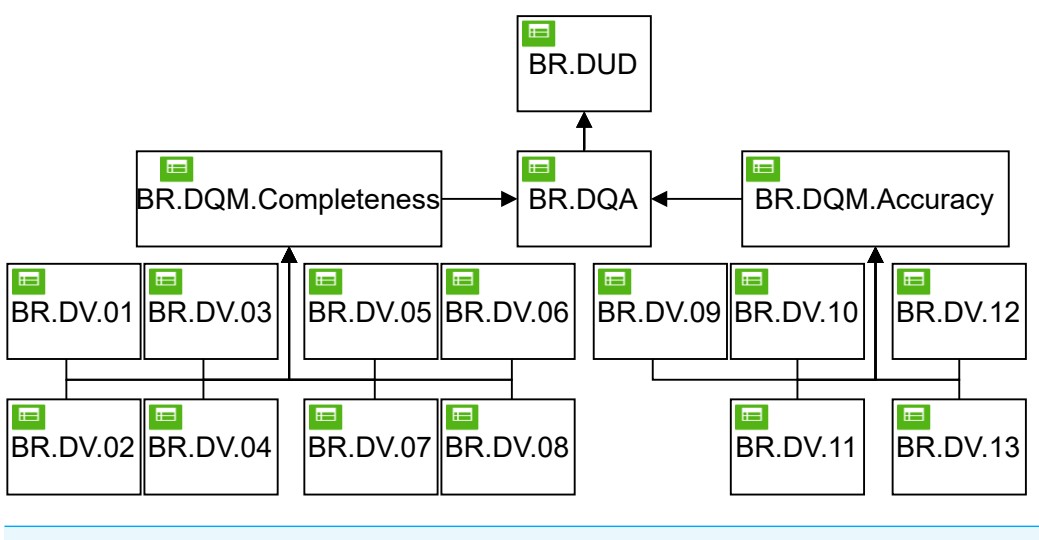

**Figure 6  Decision model diagram for DMN4DQ.**

and 45 degrees; "unusual" in case the temperature is below 1 or above 45; "unusual" will also be in case the humidity is above 0.700; and "unrealistic" in any other case.

### Define business rules for data quality measurement (BR.DQM)

Once the validation of the data from the IoT sensors has been obtained at the first level of the DMN4DQ model, the output provided by the rules activated at this level serves as input to the next level of the hierarchy, where the data quality measurement is carried out. To perform the **completeness** dimension measurement, as shown in table BR.DQM. Completeness in Fig. 7, the output of this rule depends on the outputs provided by rules BR.DV.01 to BR.DV.08. The policy applied in this rule is cumulative (C+) where for each hit a value of 1 is added, a value that coincides with a reading considered complete for each depth. Thus the higher the resulting output value, which will range from 0 to 5, the more valid information will be collected by each sensor.

Regarding the **accuracy** dimension (Table BR.DQM.Accuracy in Fig. 7), the policy applied in the rule is also cumulative (C+), in which are taking as input the outputs from the rules of the lower level (from BR.DV.09 to BR.DV.13). For each row of the rule that meets the conditions, it will add a value of 20. In this way, the measurement value of the precision dimension will be represented by a percentage from 0 to 100%, so that the more data classified as "realistic", the higher the percentage obtained at the output of this rule.

### Define business rules for data quality assessment (BR.DQA)

The outputs obtained from the data quality measurements enable the assessment of the data quality. For this purpose, a DMN table is defined (see Table *BR.DQA* in Fig. 7). The output could be:

- It will be 'suitable quality" when the completeness of the data has obtained a value greater than or equal to 5 and its accuracy is 100, *i.e.,* the reading made by the sensor is complete and accurate as all sensors obtained valid readings.

**BR.DV.04**

| F | Input VW30cm String | T30cm String | Output BR04 Number |
|---|---|---|---|
| 1 | null | null | 0 |
| 2 | null, "", "NA" | null, "", "NA" | 1 |
| 3 | - | - | 2 |

**BR.DV.09**

| F | Input VW30cm String | T30cm String | Output BR09 String |
|---|---|---|---|
| 1 | [0.150..0.700] | [1..45] | "realistic" |
| 2 | - | <1,>45 | "unusual" |
| 3 | >0.700 | - | "unusual" |
| 3 | - | - | "unrealistic" |

**BR.DQM.Completeness**

| C+ | BR01 Boolean | BR02 Boolean | BR03 Number | BR04 Number | BR05 Number | BR06 Number | BR07 Number | BR08 Number | Completeness Number |
|---|---|---|---|---|---|---|---|---|---|
| 1 | true | true | true | >=2 | - | - | - | - | 1 |
| 2 | true | true | true | - | >=2 | - | - | - | 1 |
| 3 | true | true | true | - | - | >=2 | - | - | 1 |
| 4 | true | true | true | - | - | - | >=2 | - | 1 |
| 5 | true | true | true | - | - | - | - | >=2 | 1 |
| 6 | - | - | - | - | - | - | - | - | 0 |

**BR.DQM.Accuracy**

| C+ | BR09 String | BR10 String | BR11 String | BR12 String | BR13 String | Accuracy Number |
|---|---|---|---|---|---|---|
| 1 | "realistic" | - | - | - | - | 20 |
| 2 | - | "realistic" | - | - | - | 20 |
| 3 | - | - | "realistic" | - | - | 20 |
| 4 | - | - | - | "realistic" | - | 20 |
| 5 | - | - | - | - | "realistic" | 20 |

**BR.DQA**

| F | Completeness Number | Accuracy Number | DQA {"suitable quality", "sufficient quality", "bad quality", "non_usable"} |
|---|---|---|---|
| 1 | >=5 | >=100 | "suitable quality" |
| 2 | >=3 | >=60 | "sufficient quality" |
| 3 | >=1 | <60 | "bad quality" |
| 4 | - | - | "non usable" |

**BR.DUD**

| F | DQA {"suitable quality", "sufficient quality", "bad quality", "non usable"} | DUD {"use", "do not use"} |
|---|---|---|
| 1 | "suitable quality", "sufficient quality" | "use" |
| 2 | - | "do not use" |

**Figure 7  Example of DMN tables in the decision model: BR.** DV.04, BR.DV.09, Completeness, Accuracy, Assessment and Data User Decision.

- It will be "sufficient quality" when the completeness of the data record is greater than or equal to 3 and its accuracy is greater than or equal to 60, meaning that 3 or 4 sensors have provided complete and accurate values;
- It will qualify as "bad quality" data when the reading has completeness greater than or equal to 1 and accuracy less than 60.
- It will be "non-usable" in any other case.

In this decision table (BR.DQA), the policy applied is the first hit (F) policy, which will cause the rule to provide the output associated with the first rule that is satisfied, ignoring the rest of the rules.

**Define business rules for the usability of data (BR.DUD)**
The last level within the decision hierarchy of the DMN4DQ model is the application of the business rule that determines whether data is valid to be used by the system or whether it should be rejected, *i.e.,* to determine the usability of the data. For this purpose, the rule is represented by Table BR.DUD in Fig. 7, which determines if the tuple is "use" when the BR.DQA is "suitable quality" or "sufficient quality". Otherwise, it will determine to reject the data usability.

The applied decision model allows the data provided by the IoT sensors to be automatically evaluated, and only those that meet a minimum quality defined in the business rules are taken into account, which, as a consequence, will make the system provide more reliable and higher quality results. On the other hand, data that do not

exceed the quality requirements established in the business rules will be rejected, thus preventing poor quality data from influencing the overall result of the system and even preventing damage that could occur due to decision-making based on unreliable or poor quality data.

# EVALUATION AND ANALYSIS

The evaluation of the case study of the section is carried out for two possible scenarios: the first one is based on a set of data provided by the sensors offline; in the second one, the scenario is adapted to be online, in which the IoT sensors take readings and provide them to the decision model in a continuous way, using the DMN4Spark tool.

## Offline scenario and result of the analysis

The dataset provided by *Gasch et al., 2017* on which the study is applied is processed offline. First, all the data sensors are processed and aggregated into a single database file using CHAMALEON as part of the Data curation stage. Afterwards, the analysis of the data quality was obtained once the DMN4DQ has been applied. All the material used for the evaluation is available at the Zenodo repository see offline directory): https://www.doi.org/10.5281/zenodo.8142846.

First of all, we are going to carry out a statistical analysis of the data. The sample size in the offline scenario is composed of 3,373,699 records, corresponding to readings from 42 different sensors with a frequency of 80,304 times on 3,348 different dates. The features related to humidity and temperature in the dataset use double values; otherwise 'NA' value is used as missing data. In particular, VW_30 cm holds 614 different values, VW_60 cm holds 592 different values, VW_90 cm holds 540 different values, VW_120 cm holds 539 different values, and VW_150 cm holds 527 different values. For temperature features, T_30 cm holds 384 different values (double value otherwise NA as fail), T_60 cm holds 235 different values, T_90 cm holds 207 different values, T_120 cm holds 242 different values, and T_150 cm holds 250 different values. Using Pearson's chi-square test, we obtain statistical data that determine the relationship between the location of the sensors and the values collected at different depths, both for humidity and temperature. When analysing the values obtained, it can be seen that the contingency coefficient is between 0.8061 and 0.8540 for temperature and between 0.8703 and 0.9324 for humidity. Similarly, it can be observed that the Lambda statistic for the case of temperature is between 0.0172 and 0.0272 and for humidity between 0.0657 and 0.1265. In all cases, a $p$-value of 0.00 has been obtained, from which it can be affirmed that there is a significant association between the location of the sensors and the temperature readings, as well as the humidity at the different depths with a confidence level of higher than 95%.

The hypothesis test concerning the difference between two arithmetic means (mu1-mu2) of samples from normal distributions is carried out: the null hypothesis (mu1-mu2 = 0.0) and the alternative hypothesis (mu1-mu2 <> 0.0). Given a sample of 10,000 observations with a mean of 0.0 and a standard deviation of 1.0 and a second sample of 10,000 observations with a mean of 0.0 and a standard deviation of 1.0, the calculated t-statistic equals 0.0. Since the $p$-value for the test is greater than or equal to 0.05, the null

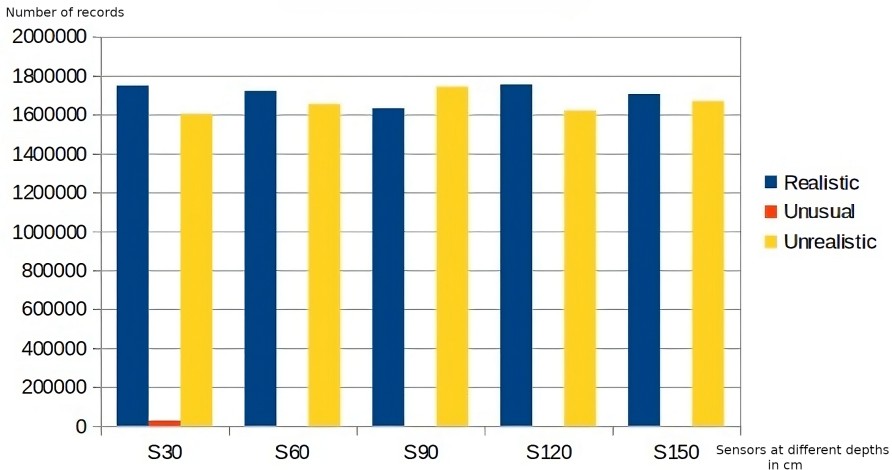

**Figure 8 Accuracy rating of data collected by sensors.**

hypothesis cannot be rejected at the 95% confidence level. The confidence interval shows that the mu1-mu2 values supported by the data fall between $-0.0277181$ and $0.0277181$. As a conclusion of the hypothesis test, we can extract that the data included in the two samples used in the test are similar, with no significant differences between the two samples.

Regarding data quality, the application of the business rules belonging to the first level of the DMN4DQ hierarchy provides the data for completeness and accuracy analysis. Figure 8 shows the results of the application of the accuracy-related business rule. On the $x$-axis, we find the sensor readings at the different depths, which as a result of the application of the business rules associated with accuracy, classify the collected data as realistic, unusual, or unrealistic. In Fig. 9, we can see how IoT sensors of different depths (S30 for 30 cm, S60 for 60 cm, etc.) identify that for sensors of 30 cm depth, there are more than 1,800,000 readings considered complete. Taking, for example, readings from sensors at 30 cm depth, more than 1,700,000 records are considered realistic when detecting humidity and temperature values within normal ranges. Unusual values are less frequent, while almost 1,000,000 records are considered unrealistic because they contain out-of-range values.

As can be seen in Figs. 8 and 9, there is a relation between the records considered realistic (accuracy dimension) and those considered as available data (completeness dimension), as well as between unavailable and unrealistic data.

Regarding the data obtained at the following levels of the decision hierarchy, Figs. 10 and 11 show the measurement of the dimensions of accuracy and completeness of the data supplied. Comparing both graphs, a correlation can be observed between the values obtained in the measurement of the accuracy dimension and the completeness dimension, where the values of poor accuracy coincide with the values with the worst assessment in completeness, as well as those readings with higher completeness (Completeness 5) have a higher accuracy value (Accuracy 100).

Continuing at the next level of the decision hierarchy, the assessment of data quality, applying the BR.DQA rules, which will combine information from both dimensions,

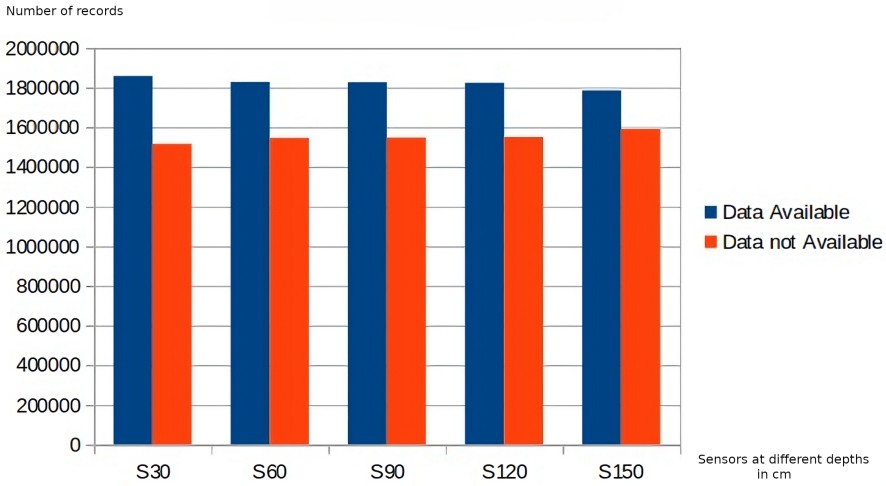

**Figure 9** **Completeness rating of data collected by sensors.**

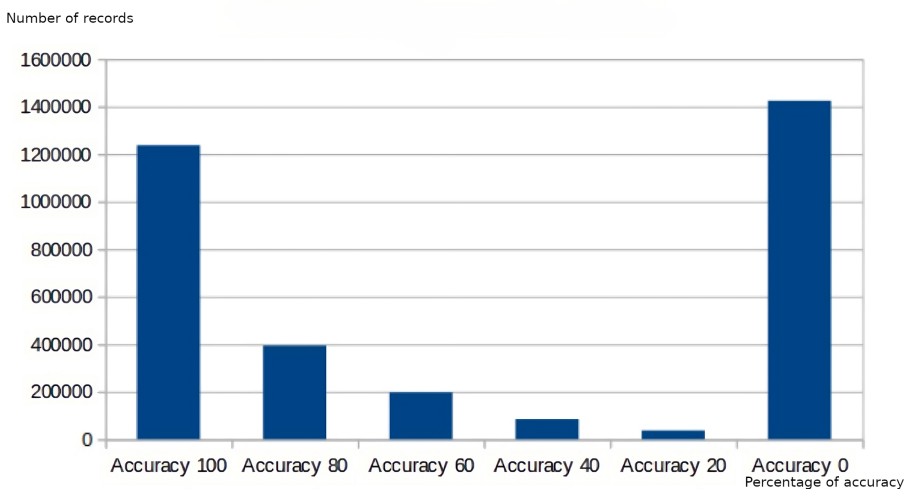

**Figure 10** **Data quality measurement for the accuracy dimension.**

completeness, and accuracy, to classify the data as adequate, sufficient, poor, or unusable quality. The results obtained have been that 42.14% is classified as bad quality data, 36.67% as suitable, 17.53% as sufficient quality, and 3.67% as bad quality.

The application of the rules contained in the last level of the hierarchy determines which data are usable and which are not, obtaining the equivalent result that 54.20% of the records can be used as they comply with the quality requirements specified by the business rules and that 45.80% of the data should not be used as they do not reach the minimum quality required.

The conclusion of the analysis of the data obtained after the evaluation of the DMN in the data quality dimensions of completeness and accuracy is that 54.20% of the data can

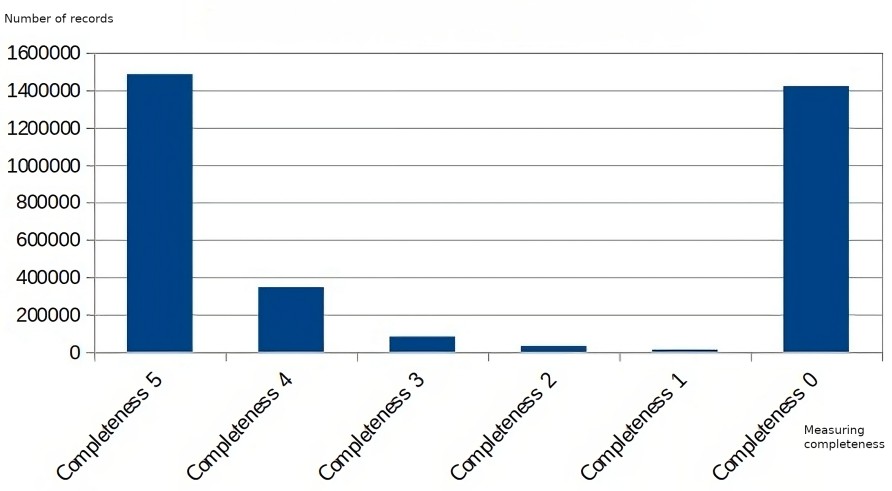

**Figure 11**   **Data quality measurement for the completeness dimension.**

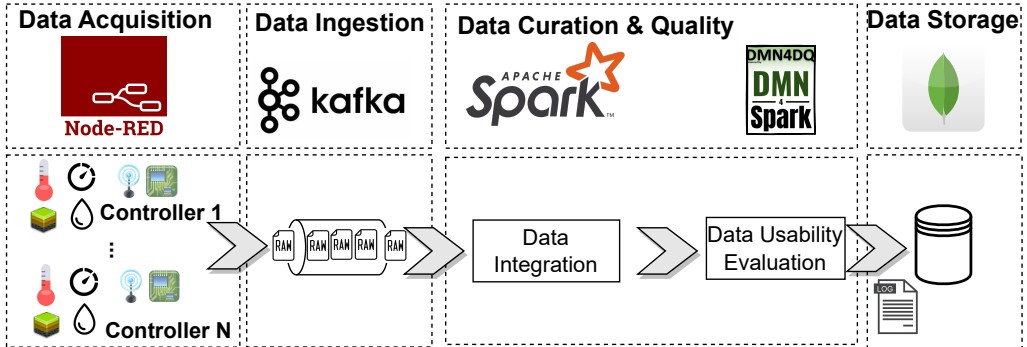

**Figure 12**   **Architecture of the streaming scenario.**

be used and that the remaining 45.80% should not be used as they contain temperature or humidity data out of range or incomplete data in an unacceptable proportion. This information can be used to make an effort to review the sensors to detect malfunctions in the operation of some sensors or even to verify the previous stages of data processing on the data classified as unusable.

## Online scenario

The objective of this section is to validate the capability of our proposal to process data in real-time. We have simulated an IoT scenario based on the same dataset used in the previous section. The scenario is composed of 50 sensors generating streaming data, producing an average of 8.5 records and 204 bytes per second. Figure 12 shows a diagram of the

architecture employed in this scenario. All the material used for the evaluation is available at the Zenodo repository (see online directory): https://www.doi.org/10.5281/zenodo.8142846. Data acquisition was simulated using Node-RED. Different factors have been included, so that the generated data includes completeness and accuracy data errors, to make the scenario as realistic as possible. These errors are produced as follows: (i) sensors might generate data with *null* values with a probability of a 3% on average; (ii) sensors might generate out-of-range data with a probability of a 5% on average; and (iii) sensors generate data at different rates, which could lead to missing values during the aggregation in time intervals.

Data ingestion is performed by Apache Kafka, being responsible for collecting the sensors' data and making it available for curation. Apache Spark consumes this data and performs the curation process according to the ELI pipeline. First, the data from the sensors must be grouped by location and by time intervals. Unlike the original dataset, which groups the data in 1-hour intervals, we decided to group the data in 30-second intervals to increase the rate of processed records. With this setting, each group was composed of an average of 33.7 records from the sensors, consuming 6.87 KB of memory per group. Once the data is grouped, the average of the measurements obtained by each sensor within each group is calculated. If any data value is not available, the result is marked as null. Once transformed, the usability of the data is calculated and stored in a MongoDB database.

We executed this scenario for 1 h. The sensors produced an amount of 43,237 data records, which were transformed into 1,292 records after grouping by location and time intervals. 593 of those records (*i.e.,* the 45%) were marked as ''use''. On the other hand, 699 (*i.e.,* the 54%) were marked as ''do not use''.

The sample size in the online scenario comprises 1,292 data records corresponding to readings generated by a total of 50 sensors in six different locations. The humidity and temperature-related characteristics in the dataset use double values; otherwise, the value ''NA'' is used as missing data. In particular, for the humidity-related data, VW_30 cm has 1,070 observations, of which 642 are different, VW_60 cm has 1,012 observations, of which 605 are different, VW_90 cm has 943 observations, of which 513 are different, VW_120 cm has 914 observations, of which 497 are different, VW_150 cm has 874 observations, of which 424 are different. Concerning the temperature data, T_30 cm has 956 different values and 1051 observations, T_60 cm has 914 different values and 995 observations, T_90 cm has 857 different values and 956 observations, T_120 cm has 848 different values and 992 observations and T_150 cm has 801 different values and 907 observations.

Similar to the offline scenario, we have carried out a statistic analysis, first, Pearson's Chi-square test is used to obtain statistical data that allow us to determine the relationship between the location of the IoT sensors and the values collected at different depths, both for humidity and temperature. In this online scenario, after analysing the values obtained, we observe that the contingency coefficient is between 0.6615 and 0.9119 for temperature and between 0.8562 and 0.9111 for humidity values. On the other hand, the Lambda statistic in the case of temperature is between 0.0755 and 0.4492, obtaining values between 0.2824 and 0.4420 for the humidity. In all cases, a $p$-value equal to 0.00 has been obtained, so we could affirm that there is a significant association between the location of the sensors and

the temperature readings as well as between the location of the sensors and the humidity with a confidence level higher than 95%.

For the online scenario we performed the hypothesis test concerning the difference between two means (mu1-m2) of samples from normal distributions. The two hypotheses tested are the null hypothesis: mu1-mu2 = 0.0, and; the alternative hypothesis: mu1-mu2 <> 0.0. Given the sample of 1,000 observations with a mean of 0.0 and a standard deviation of 1.0, and a second sample of 1,000 observations with a mean of 0.0 and a standard deviation of 1.0, the calculated Z-statistic is equal to 0.0. Since the *P*-value for the test is 1.0 (greater than 0.05), the null hypothesis cannot be rejected at the 95.0% confidence level. The confidence interval shows that the mu1-mu2 values supported by the data fall between −0.0876524 and 0.0876524. The calculated Z-statistic is 0.0. As a conclusion of the hypothesis test, we can extract that the data included in the two samples used in the test are similar, with no significant differences between the two samples.

Regarding the performance, with these settings, and during the time the scenario was run, a total amount of 8,882.24 KB of memory was consumed. The dataset that has been produced is composed of 1,292 data records, which includes the aggregated data and the results of the evaluation of the data usability, totalling 968 KB of data. Regarding the execution time, the evaluation of the usability of the data took 1.08 milliseconds for each aggregated data record. This implies that, in total, the evaluation of the usability of the whole dataset took 1,395.36 milliseconds. For the development of the experiment, a computer with a main memory of 16 GB and an Intel Core i7-1165G7 processor with 4 cores at 2.8 GHz was used, where Spark consumes the maximum allowed for each of the CPUs of the nodes in the cluster, in this case the maximum allowed for each of the four execution threads.

In conclusion, this scenario demonstrates that the proposed ELI pipeline is also applicable to IoT scenarios in which data is continuously generated, allowing the evaluation of the usability of data in real-time.

## Discussion

In both offline and online scenarios, the ability to classify data using the ELI pipeline has been tested. In the case of the offline scenario, on a dataset of 3,373,699 records that have been treated and obtaining a data curation with the determination that 54.20% of the data are likely to be incorporated into the system by classifying them as usable, disregarding the rest.

In the case of the online scenario, on the other hand, data were obtained by simulation, generating data in real-time on 50 sensors and introducing a predefined error rate in the generation of data so that both samples were similar. In this case, 43,237 records were obtained, of which 45% were classified as usable, a lower percentage than in the offline scenario.

In both cases, the data quality process worked correctly on the dataset provided, applying the business rules implemented by DMN. Obtaining a higher rate of data classified as usable depends on the characteristics of the data supplied at the input, on which the different rules of the decision tree will be applied.

According to the process performance, the data quality assessment does not depend on if the data have come from an offline or online scenario. The main difference is the curate phase, where the data integration can be done online, or it must be executed online with reads of the sensor in windows of time. In relation to the evaluation, neither solution implies a technological challenge; the difficulty is that in an online scenario, only a subset of the dataset is considered, and the curation step is executed online, integrating the various sensor reads in a tuple to be assessed. Depending on the size of the window time, different tuples could be created and assessed.

## CONCLUSIONS AND FUTURE WORKS

There are several challenges facing IoT scenarios, especially heterogeneous data sources that collect data from the environment and inject it into the system, which must be treated more or less homogeneously. For this reason, it is necessary to perform the data curation process and determine the quality of data so that only data that meets minimum requirements are accepted. Both processes allow the overall operation of the system and the business logic to be the most appropriate, discarding poor-quality data that can contribute little to the system except in negative aspects, such as incorrect decision-making or the introduction of errors. We have defined an IoT-aware Big Data Pipeline, which integrates CHAMALEON and DMN4DQ to facilitate data curation and data quality aspects. The use of CHAMALEON facilitate the data transformation in a single dataset, meanwhile, DMN4DQ provides users with a method to measure and assess data quality. Further, we have integrated and demonstrated functionality in offline and online scenarios. Thus, both CHAMALEON and DMN4DQ have been used to transform, measure and evaluate the quality of all the data coming from IoT sensors in a real scenario, receiving as output the recommendation in terms of the usability of the data in the defined quality dimensions (completeness and accuracy). After applying the proposed model, the result was that 54% of the data collected by IoT sensors in the offline scenario met the defined quality requirements, and the rest of the data did not meet the same requirements, so that they could be discarded. In the online scenario, 45% of the data were selected as usable. In both cases, the amount of data rejected for being of low quality is relevant, and this fact should be taken into account. Among the benefits obtained by applying the model is the fact that only quality data are selected, providing greater reliability and trustworthiness in the data provided after applying the data curation process. However, data that do not reach a minimum quality after being evaluated are discarded and no longer need to store these data, or we can choose to analyse data problems and try to find corrective actions to "fix" the problems encountered.

Among the limitations of the proposed approach are the following: (1) the DMN model must be known to implement the business rules that deal with data from IoT sensors; (2) only two dimensions have been considered in the validation of the model: completeness and accuracy, which could be insufficient in different scenarios and require greater complexity.

Regarding future work, the possibility of integrating a data analysis model within ELI that allows statistical analysis compatible with the hierarchy of the DMN business

rules is considered, as well as the incorporation of security measures that guarantee data privacy during the data quality analysis process. Other future lines of research would be the integration of the selection of the corrective actions to fix data quality problems detected and the integration of the data curation process in combination with blockchain technology, where to take advantage of the benefits of this technology.

### Funding
This work has been funded by the projects AETHER-US (PID2020-112540RB-C44/AEI/10.13039/501100011033) and ALBA-US (TED2021-130355B-C32) by MCIN/AEI/10.13039/501100011033, COPERNICA (P20_01224) and METAMORFOSIS (US-1381375). The funders had no role in study design, data collection and analysis, decision to publish, or preparation of the manuscript.

### Grant Disclosures
The following grant information was disclosed by the authors:
AETHER-US: PID2020-112540RB-C44/AEI/10.13039/501100011033.
ALBA-US (TED2021-130355B-C32): MCIN/AEI/10.13039/501100011033.
COPERNICA: (P20_01224).
METAMORFOSIS: US-1381375.

### Competing Interests
The authors declare that there are no competing interests.

### Author Contributions

- Francisco José de Haro-Olmo conceived and designed the experiments, performed the experiments, analyzed the data, performed the computation work, prepared figures and/or tables, authored or reviewed drafts of the article, and approved the final draft.
- Alvaro Valencia-Parra conceived and designed the experiments, performed the experiments, analyzed the data, performed the computation work, prepared figures and/or tables, authored or reviewed drafts of the article, and approved the final draft.
- Ángel Jesús Varela-Vaca conceived and designed the experiments, performed the experiments, analyzed the data, performed the computation work, prepared figures and/or tables, authored or reviewed drafts of the article, and approved the final draft.
- José Antonio Álvarez-Bermejo conceived and designed the experiments, performed the experiments, analyzed the data, performed the computation work, prepared figures and/or tables, authored or reviewed drafts of the article, and approved the final draft.
- María Teresa Gómez-López conceived and designed the experiments, performed the experiments, analyzed the data, performed the computation work, prepared figures and/or tables, authored or reviewed drafts of the article, and approved the final draft.

### Data Deposition
The raw data and code are available at Zenodo: ajvarela. (2023). ajvarela/eli-iot-bigdata-pipeline: New online results (v.1.1.0). Zenodo. https://doi.org/10.5281/zenodo.8142846.

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
