# Peer review of "ELI: an IoT-aware big data pipeline with data curation and data quality"

_PeerJ Computer Science, doi:10.7717/peerj-cs.1605_

## Round 0.1 · original submission · Major Revisions

Please, consider all the reviewers comments.

On the other hand, statistical analysis needs to be accompanied by methodological information to prove its validity. Specifically, some parameters must be specified, such as the sample size, the appropriate statistical measures, multiple testing correction, effect size, degrees of freedom, and reporting of relevant figures such as statistic, n, and exact p-values.

Reviewer 1 ·

Basic reporting

The paper presents a proposed pipeline for data quality assessment and curation for use in IoT/big data applications.

Although I find the proposal very interesting, several questions arise:

My main concern is that the contributions are not clear. It is not clear what progress this paper has made with respect to the authors' previous work: (DMN4DQ: When data quality meets DM, DOI:10.1016/j.dss.2020.113450; and Data curation in the Internet of Things: A decision model approach; DOI: 10.1002/cmm4.1191).

Besides, it would be appreciated to have introduced the terms DMN, not only the explanation of the acronyms (used in the abstract and introduction without prior definition), but in particular the concept and the relation to IoT and big data and of course to explain it before explaining DMN4DQ. It definitely would improve the understanding of the paper from the very beginning.

Experimental design

Concerning the case study, which seems to be the major contribution (though it was partly published in one of the references before), it raises some doubts. For example, it is not clear what the difference between empty, null or without assigned value explained in line 247 is.

Furthermore, why does completeness take values from 0 to 5 and accuracy from 0 to 100; are these measures arbitrary, are they based on some standard or convention, how do you know for accuracy what is realistic, unusual or unrealistic, is it subjective, is it susceptible to a margin of error, and what is the authors' basis for deciding what is realistic, unusual or unrealistic? On what basis do the authors decide that greater than 3 is sufficient quality and 1 or less is bad quality? Is there a procedure or methodology followed to establish these categories or is it all arbitrary and subjective to the person deciding? Is there a common basis for all case studies or can each case study have different levels? With all these doubts it is difficult to give value to Figures 7 to 10 and to the proposal in general, in summary it requires better motivation and explanation.

On the other hand, at the end of the case study it is specified that Apache Spark consumes the data and performs the curation according to the methodology. It would be relevant to know the resource consumption, the efficiency of the process and the latency to obtain the classification.

Validity of the findings

no comment

Additional comments

Minor issues:

The way references have been included should be revised, some punctuation is missing to separate them from the text ( for example in line 39 "data quality, privacy, and security Choi et al. (2017)").
Also sometimes the same issue for references to figures (e.g. in line 320 "As can be seen in figures 7 8 there is ....").

Finally, it is not clear to me whether the lines of the rectangles in Figure 2 have not been made straight for some special purpose.

Cite this review as

·

Basic reporting

This study presented an IoT-based Big Data pipeline that allows to assess the usability of the data collected by IoT sensors in offline and online scenarios. The proposed approach integrates a data curation process where data can be measured and evaluated according to several data quality dimensions.

However, there are some comments that need to be addressed in this manuscript.

The "Abstract" section can be made much more impressive by highlighting your contributions. The contribution of the study should be explained simply and clearly. What is your contribution exactly?
It is better to add numerical results to show the effectiveness of your proposed methodology compared to the most recent state-of-the-art.
The main steps of the proposed methodology (pipeline) need more clarification, and what are the main differences between the proposed one and the standard machine learning pipeline?

Data curation in the proposed methodology is not clear. How do the authors satisfy this stage in big data?

The authors  utilize the DMN4DQ methodology in  big data and need more highlighting to show the importance of this technology.

It is better to add more details to the decision model diagram "DMN4DQ" to give the reader a good view of your methodology.
The abbreviations in the study are ambiguous. It is better to give the full name of the abbreviations before using them in your manuscript.

Experimental design

The assessment of the proposed pipeline is not clear. How did you compute the accuracy?

Validity of the findings

A discussion section is needed to be added to show the findings from the offline and online scenarios.

---

## Round 0.2 · Minor Revisions

One of the reviewers has indicated that some changes are necessary for your paper to be published in PeerJ Computer Science journal. Please, consider making the comments indicated by the reviewer and resubmitting the paper to be considered for publication.

Emilia

Reviewer 1 ·

Basic reporting

The main contributions of the paper are not clear yet, I guess your contributions should be understood by the reader when reading the following paragraph In the introduction: “In summary, the proposal is an integration of data curation and data quality assessment in a single big data pipeline, being a combination of the previous processes in a holistic solution. Furthermore, we enlarge the data quality assessment to online scenarios, as IoT environments require this type of analysis that was not included in previous solutions”; however, you should clearly state there which are your contribution in this paper, making it clear what it was already published and which are the new findings in this paper.

Experimental design

Concerning the resource consumption, is not only relevant memory and latency but also CPU consumption, and it is also necessary to know the features of the machine used for the tests in order to be able to assess performance.

Validity of the findings

Please include in every chart figure what axis x and y represent, and their units, when necessary.

Additional comments

Finally, I found CHAMALEON, CHAMELEON, CHAMELON along the paper, is it all the same thing? Please correct or explain it.

Cite this review as

·

Basic reporting

Authors have improved the paper in this revision.
The manuscript can be accepted in its current form.

Experimental design

Authors have improved the paper in this revision.
The manuscript can be accepted in its current form.

Validity of the findings

Authors have improved the paper in this revision.
The manuscript can be accepted in its current form.

Additional comments

Authors have improved the paper in this revision.
The manuscript can be accepted in its current form.

---

## Round 0.3 · accepted · Accept

Thank you for considering the reviewers' comments and making the required changes to your article.

I am happy to inform you that your paper has now reached the required level for publication in PeerJ Computer Science. Thank you for considering our journal for publishing your research papers.

We hope you will continue to consider the journal for publication in your future research.

Sincerely,

M. Emilia Cambronero
Academic Editor of Peer J Computer Science